# Agreement between Inertia and Optical Based Motion Capture during the VU-Return-to-Play- Field-Test

**DOI:** 10.3390/s20030831

**Published:** 2020-02-04

**Authors:** Chris Richter, Katherine A. J. Daniels, Enda King, Andrew Franklyn-Miller

**Affiliations:** 1Sports Medicine Research Department, Sports Surgery Clinic, Santry Demesne, Dublin 9, Dublin, Ireland; KatherineDaniels@sportssurgeryclinic.com (K.A.J.D.); endaking@sportssurgeryclinic.com (E.K.); afranklynmiller@me.com (A.F.-M.); 2Queen’s School of Engineering, University of Bristol, BS8 1QU, UK; 3Centre for Health, Exercise and Sports Medicine, University of Melbourne, VIC 3053, Melbourne, Australia

**Keywords:** optical motion capture, inertial motion capture, field test, movement analysis

## Abstract

The validity of an inertial sensor-based motion capture system (IMC) has not been examined within the demands of a sports-specific field movement test. This study examined the validity of an IMC during a field test (VU®) by comparing it to an optical marker-based motion capture system (MMC). Expected accuracy and precision benchmarks were computed by comparing the outcomes of a linear and functional joint fitting model within the MMC. The kinematics from the IMC in sagittal plane demonstrated correlations (r*^2^*) between 0.76 and 0.98 with root mean square differences (RMSD) < 5°, only the knee bias was within the benchmark. In the frontal plane, r*^2^* ranged between 0.13 and 0.80 with RMSD < 10°, while the knee and hip bias was within the benchmark. For the transversal plane, r*^2^* ranged 0.11 to 0.93 with RMSD < 7°, while the ankle, knee and hip bias remained within the benchmark. The findings indicate that ankle kinematics are not interchangeable with MMC, that hip flexion and pelvis tilt higher in IMC than MMC, while other measures are comparable to MMC. Higher pelvis tilt/hip flexion in the IMC can be explained by a one sensor tilt estimation, while ankle kinematics demonstrated a considerable level of disagreement, which is likely due to four reasons: A one sensor estimation, sensor/marker attachment, movement artefacts of shoe sole and the ankle model used.

## 1. Introduction

Motion capture systems are frequently used to assess athletic performance or to support return to play decisions, as they provide an objective insight into the kinematics of athletes, which can be compared with, either previously captured data or other benchmarks (e.g., a normative group). A commonly used technology to capture movements is marker-based optical motion capture (MMC) systems—Systems that combine cameras and active or passive markers. MMC systems record the positions of reflective markers that are attached to an object and are considered to be state of art in motion capture, due to their high accuracy when tracking rigid objects [1]. However, MMC systems are bound to a capturing volume, often requiring considerable manual post-processing, and are sensitive to disturbances of the recording cameras and infra-red interference from other sources. As a consequence, MMC systems are often found only in research laboratories. A limitation of capturing movements, within a laboratory setting, is that the assessed individual is fully aware of the assessment and there is a lack of external distraction (e.g., opponents) due to the restricted capturing volume. Another technology that can be used to capture movements is inertial-based motion capture (IMC). IMC systems are not bound to a capturing volume, do not require manual post-processing and allow out-of-laboratory testing with rapid results output [2]. An IMC system uses multiple inertial sensors that combine multiple devices within a unit (one or more accelerometers, gyroscopes and magnetometers), and estimate position and orientation by inferring change of position relative to a past time-point by double-integrating the data. This process, however, is susceptible to drift – the accumulation of small measurement errors within the sensor over time. To remove/reduce the effects of the sensor drift, Kalman filters or gradient descent optimization algorithms are commonly used [3]. However, because of the drift in the inertial sensors, the accuracy of IMC systems is somewhat compromised. Previous studies, which validated IMC against MMC systems, reported good agreement in pelvis sagittal (root mean square difference [RMSD] = 8.89; r*^2^* > 0.74) and frontal (RMSD = 4.44; r*^2^* > 0.74) plane angles during daily living activities (gait, sit-to-stand transfers and block step-up) [4], trivial to small differences in sagittal pelvis and lower extremity kinematics during kicking [2] and very high validity (r*^2^* > 0.80) in the sagittal plane of hip, knee, and ankle joint angles during walking, jumping and squatting, but only demonstrating acceptable validity (0.40 < r*^2^* < 0.08) in the frontal and transverse planes for squatting and jumping [5]. However, while these studies imply that IMC systems can be used to record movements, the duration of the activities examined is relatively short (<2 s), and higher measurement errors would be expected to be observed in longer data captures [6]. The ability to capture movement accurately, over a more-extended period, enables the analysis of movement sequences, which potentially increases the external validity as preparation time and internal conscious focus on the executed movement tasks reduce. Practitioners currently utilize field-based tests of acceleration, deceleration and change of direction for this purpose, aiming to simulate the demands of a real-time sporting scenario by incorporating a series of tasks that have to be performed in the shortest time possible. The practical use of MMC systems to record data from these tests is limited because of the following three shortcomings: A large capture volume is needed, obstruction of marker visibility and consequently high post-processing demands. In contrast, an IMC system does not suffer these three short-comings and offers a solution in enabling efficient feedback of objective movement metrics to athletes and coaches. However, the validity of an IMC system, during a field test or game-like situation, is not currently known.

The aim of this study is to examine the accuracy and precision validity of an IMC system during a field test.

## 2. Materials and Methods

### 2.1. Subjects

Six male recreational athletes participated in this study. All athletes consented to the study. Ethical approval was granted from the Sports Surgery Clinic ethics committee.

### 2.2. Field Test

The chosen field test (VU®; Sports Surgery Clinic, Dublin) was developed by clinicians for use within their clinical practice, and used as a field test as it simulates athletic qualities associated with a variety of field sports, due to phases of acceleration, deceleration, change of direction, curved and backwards running, as well as a 15-meter sprint. These features are often assessed independently (e.g. The T test, The Change-of-Direction and Acceleration Test, The Illinois agility run, the pro agility test, the 3-cone drill, the L run or the 505), but have not previously been combined in a single test. The layout of the test is illustrated in Figure 1 and the flow of the test is described in the figure caption.

### 2.3. Data Capture

Participants were tested twice and asked to complete 10 VU® tests (alternating between left- and right-sided) within each session. Data capture was performed at an indoor motion capture facility (Audio-motion Studios Limited, Oxford, UK) that was able to accommodate optical motion capture within the volume needed. This motion capture studio was equipped with 70 cameras that recorded a capturing frequency at 200 Hz and were mounted on scaffolding.

At the start of each session, each participant was asked to perform a self-selected warm-up and was subsequently familiarized with the field test by performing two walking and two jogging effort trials. After familiarization, eight inertial measure units (IMC sensor; 100 Hz; MTw2; Xsens Technologies B.V, Enschede Netherlands) were attached to the participant using Velcro straps, as per Xsens recommendations. To record the kinematics of the trunk and pelvis, IMC sensors were placed on the upper part of the sternum and directly over the sacrum, using double-sided adhesive and zinc oxide tape to keep the IMC sensor in position during data capture. To record the kinematics of the left and right limb, two IMC sensors were was placed between the layers of the Velcro strap, to secure its position and minimize any movement, the first was placed centrally on the upper limb halfway, between the greater trochanter and lateral epicondyle of the femur (thigh sensor) and medial on the proximal surface of the tibia (shank sensor). To record kinematics of the left and right foot, an IMC sensor was placed on the dorsum of each foot using double-sided adhesive tape, as well as zinc oxide tape. After the IMC sensors were placed, 32 retroreflective markers (14 mm diameter) were attached to the participants using double-sided adhesive tape as well as zinc oxide tape to keep retroreflective markers in position during the data capture. Markers were attached so that both the Plug-In-Gait (PiG) model and the optimal shape symmetrical centre of rotation estimation approach (OSSCA) could be used. 

After the IMC sensors and markers were placed a “reference” trial was performed to calibrate the motion model - range of motion, as per Vicon for the MMC and n-pose walk [7] for IMC. During the range of motion reference trial, the participant stood in a stationary position, and then for each leg, sequentially flexed/extended the knee, flexed/extended the hip, abducted/adducted the hip and circum-ducted the hip, as per Vicon recommendations. During the walking reference trial, the participant stood in a stationary position, walked 2 m forwards in a straight line, turned around, walked back to the starting position and turned around to face the original heading direction. Subsequently, participants were asked to complete a maximal effort trial on either, the left or right side (randomized), rest for 2 min and perform a maximal effort trial on the opposite side. This was done until a total of 10 trials were captured. If a marker fell off the investigators re-attached the marker and repeated the reference range of motion trial. If an IMC sensor fell off, the investigators re-attached the sensor and repeated the reference walking trial. The data capture was started independently in the optical and IMC system.

### 2.4. Data Preparation

To prepare the data for analysis, the following steps where performed: Data post-processing, joint angle calculation, system synchronization and data extraction (Figure 2). Data post-processing and joint angle calculation were performed first for both the IMC and the MMC, as the centre of mass (CoM) measures were used for synchronisation, which was in turn needed for the event detection.

The IMC data was processed using MVN Analyze 2019.2 (Xsens Technologies B.V, Enschede Netherlands) and the data was then exported as mvnx files. Processing included filtering, skin movement artefact reduction and functional joint estimation via the MVN Fusion Engine. To calculate joint kinematics, the processed quaternion data of the trunk, pelvis, thigh, shank and foot were exported from the mvnx file and used to calculate the rotation matrix (distal to proximal) between trunk and pelvis (thorax angle), pelvis and thigh (hip angle), thigh and shank (knee angle), and shank and foot (ankle angle), which was subsequently transformed into Euler angles using a Y-X-Z sequence. The pelvis angle/pelvis quaternion was expressed on the external reference trial, computed using the t-pose and then transformed into Euler angles using a Z-Y-X sequence. The CoM in x, y and z as well as the foot contact definitions (as generated by the MVN Fusion Engine) were also exported for further processing. Joint kinematics, CoM measures and foot contact definitions were then exported into a SQL file.

After processing the IMC data, the optical motion data were processed. Processing included the reconstruction of marker positions, marker labelling, filling of gaps in marker trajectories, filtering of marker trajectories (Woltring; 15 Hz cut-off [8]) and the fitting of the segments to marker trajectories. Gap filling was performed using a segment fill approach if possible and a pattern fill or spline fill approach if not. Segment bones were then fitted to the marker trajectories, using the PiG pipeline as per Nexus 2.8.1 (Vicon Metrics Group Ltd, Oxford, UK) and the OSSCA pipeline [9] as per the Nexus Advanced Gait Workflow (Nexus 2.8.1, Vicon Metrics Group Ltd, Oxford, UK). OSSCA is a workflow that combines the optimal common shape technique (OCST; [10]), symmetrical axis of rotation approach (SARA; [11]) and symmetrical centre of rotation estimation (SCoRE; [12]), and seeks to minimize the sensitivity of joint kinematics to marker placement. The hip joint centre and knee and ankle axes were then calculated using the symmetrical centre of rotation estimation (SCoRE; [12]), and the symmetrical axis of rotation approach (SARA; [11]), respectively. Soft tissue artefact was minimized using the optimal common shape technique (OCST: [10]), where an optimum rigid marker configuration for each segment is formed to reduce the effects of skin elasticity. To calculate joint kinematics for the MMC, the origin of the segment data from the trunk and pelvis (thorax angle), pelvis and thigh (hip angle), thigh and shank (knee angle), as well as shank and foot (ankle angle) was translated into the global origin and then used to compute a rotation matrix (distal to proximal) that was subsequently transformed into Euler angles using a Y-X-Z sequence, for the outcomes of PiG, and OSSCA, respectively. The pelvis angle was calculated by transforming the rotation matrix between the global coordinate system and a translated pelvis segment (origin in global origin) using a Z-Y-X sequence. The x, y and z global coordinates of the CoM position was also exported. Joint kinematics and CoM measures were then exported into a SQL file.

To synchronize the IMC and MMC, the IMC data was up-sampled to 200 Hz. The vertical position of the CoM was then extracted from the SQL file for both systems and used to compute CoM vertical velocity. The timing offset between the systems was then defined by the lag index that corresponded to the maximal correlation coefficient observed during a cross-validation and subtracted from the device that triggered data capture first. To ensure both data sets contained the same number of samples, both data sets were cropped to the time point, 0, and the minimum duration across the systems. The last step in processing was the rotation of the horizontal IMC CoM measures in the CoM description of the MMC (*x* = left-right direction; *y* = forward-backward) using a rigid transformation.

The final step in the data preparation was event detection (Figure 3). The field test can be separated into the following phases: Hurdle hop (HuHo), the first change of direction (V1) phase, backward running (Backwards) phase, p the second change of direction (V2) phase, the first curved run (U1) phase, the second curved run (U2) phase and the sprint (Sprint). For a trial that started with a left limb lateral HuHo, the start of the HuHo was identified as the second impact of two subsequent contacts of the left limb (where vertical CoM velocity was below 2 m per second). The end of the HuHo was defined as the toe off within that contact. The start of the V1 phase was defined as the frame following the end of the HuHo and ended at the local next local minima of the CoM x. The backward running phase started at the frame, following the end of the V1 phase and ended at the next local maxima of CoM x. The V2 phase was defined to start at the frame, following the end of the Backward phase and ended at the next local maxima of CoM y. The U1 phase was defined to start at the frame following the end of the V2 phase and ended at the next local maxima of CoM y. The U2 phase was defined to start at the frame, following the end of the U1 phase, and ended at the next local minima of CoM y. The Sprint phase was defined to start at the frame following the end of the U2 phase and ended when the CoM y passed the 20-meter mark.

### 2.5. Statistical Analysis

The reconstruction of three-dimensional (3D) marker positions using optical motion capture is reliant on simple triangulation and is accurate to <1 mm [1]. However, multiple approaches (models) calculate the tri-planar joint angles from these marker position trajectories [12,13,14,15,16]. These models differ in degrees of freedom and constraints, applied at each joint, as well as compensation for skin movement artefacts and calculation of joint centres and axes of rotation. This can result in considerable differences in outcome measures between different models, and it is therefore, not possible to compare IMC-derived kinematics against a single universal MMC-based ‘gold standard’. For the purpose of this study, we defined the OSSCA models as most appropriate and as a reference standard. To interpret the magnitude of differences (bias) between OSSCA derived kinematics and IMC outcomes, a benchmark was computed by comparing kinematics derived from OSSCA and the commonly used PiG model. Both accuracy (systematic [device] and random error) and precision (random error) metrics were calculated, in order to differentiate systematic and random errors in the IMC system. If differences between IMC and MMC were no larger than differences between OSSCA and PiG, we concluded that the accuracy/precision of the IMC system for the relevant variable was acceptable, e.g., within the error resulting from differences in MMC model assumptions. 

Both accuracy and precision were examined by using integrated pointwise indices, within each phase of joint kinematics and horizontal CoM measures [17], and also for the reference trial. The accuracy of the IMC system was expressed based on thhe level of agreement between the OSSCA and IMC outputs, by calculating the bias (differences) and 95% limits of the bias [18]. The precision of the IMC system was expressed by the root mean square prediction difference (RMSD) of a regression analysis [19], between the OSSCA and IMC outcomes, the % RMSD ([RMSD/OSSCA signal] * 100) and the resulting squared correlation coefficient (r^2^; Pearson). The r*^2^* was used to examine shared variation and classified into very high (1.00 > r^2^ > = 0.81), high (0.81 > r^2^ > = 0.49), moderate (0.49 > r^2^ > = 0.25), low (0.25 > r^2^ > = 0.09), negligible (r^2^ > 0.09) [20]. No test for statistical differences was performed as accuracy and precision measures were compared to the benchmark set by the PiG vs. OSSCA comparison. Limits of agreement plots were visually screened for proportional bias. Bias, 95% limits of the bias, RMSD, % of RMSD and r^2^ were stored in a phase (n = 7) × observation (n = 120 [10 VU × 6 subjects × 2 assessments]) matrix and are reported as the mean and standard deviation of each phase and the overall average of the mean and standard deviation.

Data processing and analysis was performed in Python 3.7.

## 3. Results

All six athletes (mean ± standard deviation age: 29.8 ± 5.0 years; height: 182.0 ± 4.2 cm; body mass: 84.0 ± 4.8 kg) completed 10 VU trials across 2 sessions, approximately 4 h apart. Two trials from subject 4 had to be excluded due to issues with the IMC system (connectivity issues in the beginning of the trial were not noticed until the trial ended), while 1 trial in subject 5 had to be excluded, as the thigh segment markers had large gaps that could not be reliably filled. Completion times (Table 1) were 19.48 ± 1.21 Section in the first session and 19.86 ± 1.60 Section in the second session (depended t-test; *p* = 0.189; r^2^ = 0.89).

### 3.1. Accuracy

In the sagittal plane, the observed mean bias (std bias, 95% limits) was −4.58 (2.98, −13.20 to 4.04), −4.11 (4.30, −14.30 to 6.09), −12.10 (6.36, −22.72 to −1.49), −16.60 (6.55, −21.27 to −11.93) and −7.58 (6.43, −14.01 to −1.15) for the ankle, knee, hip, pelvis and trunk joint/segment kinematics from start to end. A detailed description of the accuracy measures in the sagittal plane is reported in Table A1. Ankle and hip kinematics bias were larger in the IMC system than the benchmark, while knee kinematics were within the benchmark (Figure 4). Hip joint kinematics were judged to have a proportional bias in 12 of the 12 sessions, with larger errors for larger magnitudes. The main bias (% overlap with PiG bias; 95% limits) observed within the reference model was in the sagittal plane was −1.55 (95%; −8.02 to 4.92), −6.25 (82%; −15.25 to 2.76), −5.16 (91%; −13.46 to 3.15), −16.27 (N/A; −23.24 to −9.30) and −15.51 (N/A; −24.17 to −6.86) for the ankle, knee, hip, pelvis and trunk joint/segment kinematic.

In the frontal plane, the observed mean bias (std bias, 95% limits) was −1.41 (5.81, −22.27 to 19.45), 9.40 (3.75, −2.19 to 20.99), −8.51 (1.98, −17.73 to 0.72), −0.58 (3.58, −5.40 to 4.24) and −1.43 (4.92, −8.03 to 5.18) for the ankle, knee, hip, pelvis and trunk joint/segment kinematics from start to end. A detailed description for the accuracy measures in the frontal plane is reported in Table A2. Ankle joint kinematics bias were higher than the benchmark. A large portion of the observations (75%) of the hip joint bias were higher than the benchmark. The knee joint bias was smaller than the benchmark (Figure 5). Ankle and knee joint kinematics were judged to have a proportional bias in 12 of the 12 sessions, with larger errors for larger magnitudes. The main bias observed within the reference model was in the frontal plane was >0.01 (100%; 0 to >0.01), 5.90 (77%; 1.15 to 10.65), −4.08 (91%; −10.05 to 1.89), −0.24 (N/A; −4.73 to 4.24) and −0.94 (N/A; −7.88 to 6.00) for the ankle, knee, hip, pelvis and trunk joint/segment kinematic.

In the transverse plane, the observed mean bias (std bias, 95% limits) was 3.23 (9.62, −23.29 to 29.75), 6.27 (7.84, −11.65 to 24.19), 9.91 (4.59, −8.96 to 28.78), 3.29 (18.17, −3.18 to 9.99) and −1.63 (5.90, −7.37 to 4.10) for the ankle, knee, hip, pelvis and trunk joint/segment kinematics from start to end. A detailed description for the accuracy measures in the transverse plane is reported in detail in Table A3. Ankle joint kinematics bias were higher than the benchmark. A large portion of the observations (~70%) of the knee joint bias were higher than the benchmark. The hip joint bias was smaller than the benchmark (Figure 6). In the ankle joint, the observed bias was the proportional in 12 of the 12 sessions, with larger errors for larger magnitudes. The main bias observed within the reference model was in the transversal plane was −15.50 (55%; −30.36 to −0.64), 11.19 (31%; −3.16 to 25.53), 8.60 (88%; −6.07 to 23.28), 0.01 (N/A; −5.35 to 5.38) and −1.23 (N/A; −8.38 to 5.91) for the ankle, knee, hip, pelvis and trunk joint/segment kinematic.

The observed mean bias (std bias, 95% limits) for the CoM was 0.01 meter (0.17, −0.18 to 0.20) and −0.07 meter (0.15, −0.28 to 0.14) for the left-right and backward-forward direction. A detailed description of the accuracy measures for the COM metrics is reported in Table A4. One of the 12 sessions was judged to have a proportional bias in the backward-forward direction, with larger errors for larger magnitudes. 

### 3.2. Precision

In the sagittal plane, the correlation between IMC and OSSCA was very high in the ankle (r^2^ = 0.89 ± 0.08, RMSD = 4.31 ± 1.46~7%), knee (r^2^ = 0.96 ± 0.02, RMSD = 4.44 ± 1.25~4%), hip (r^2^ = 0.97 ± 0.03, RMSD = 3.21 ± 0.95~4%) and pelvis (r^2^ = 0.84 ± 0.13, RMSD = 2.09 ± 1.06~9%), and high in the trunk kinematics (r^2^ = 0.77 ± 0.16, RMSD = 2.89 ± 1.38~10%). A detailed description of the precision measures in the sagittal plane is reported in Table A1. Ankle kinematics RMSD were higher in the IMC system than the benchmark, while RMSDs of the knee were within the benchmark. Hip joint bias was for more than half of the observations in the benchmark (Figure 7). In respect to the r^2^ benchmark: Ankle joint correlations were below the benchmark, knee and hip joint were within or slightly below (Figure 7).

In the frontal plane, the correlation between devices was low in the ankle (r^2^ = 0.19 ± 0.13, RMSD = 9.31 ± 1.38 ~ 20%) and knee kinematics (r^2^ = 0.14 ± 0.16, RMSD = 3.39 ± 0.88 ~ 13%) and high in the hip (r^2^ = 0.64 ± 0.26, RMSD = 3.76 ± 1.88 ~ 13%), pelvis (r^2^ = 0.79 ± 0.15, RMSD = 2.07 ± 0.79 ~ 10%) and trunk kinematics (r^2^ = 0.75 ± 0.18, RMSD = 2.44 ± 0.92 ~ 12%). A detailed description of the precision measures in the sagittal plane is reported in Table A2. Ankle kinematics RMSD were much higher in the IMC system than the benchmark. Hip kinematics RMSD were higher in about 50% of the observations, while knee RMSDs in the lower parts of the benchmark (Figure 8). With respect to the r^2^ benchmark: Ankle and hip joint correlations were below the benchmark, the knee joint (even with low correlations) was on the lower ranged of the benchmark (Figure 8).

In the transverse plane, the correlation between devices was low in ankle (r^2^ = 0.12 ± 0.12, RMSD = 6.09 ± 1.55~20%) and knee kinematics (r^2^ = 0.20 ± 0.13, RMSD = 3.18 ± 0.61~17%), moderate in the hip (r^2^ = 0.25 ± 0.20, RMSD = 4.68 ± 1.39~18%) and very high in pelvis (r^2^ = 0.92 ± 0.09, RMSD = 2.86 ± 2.34~5%) and trunk kinematics (r^2^ = 0.86 ± 0.10, RMSD = 2.36 ± 0.97~9%). A detailed description of the precision measures in the sagittal plane is reported in Table A3. The ankle joint RMSD were within, but in the upper parts, of the benchmark. The knee RMSD were in the lower ranges of the benchmark and the hip joint RMSD were within the benchmark (Figure 9). With respect to the r^2^ benchmark: Ankle and knee joint correlations were below the benchmark, the hip joint were correlations were within the benchmark (Figure 9).

The correlation between devices for the CoM was very high for the axis capturing movements in the left-right direction (r^2^ = 0.98 ± 0.03, RMSD = 0.05 ± 0.03~2%) and the backward-forward direction (r^2^ = 0.94 ± 0.08, RMSD = 0.07 ± 0.04~4%). A detailed description of the precision measures for the center of mass measures is reported in Table A4. 

## 4. Discussion

This study examined the accuracy and precision of lower-limb kinematics, quantified using an inertia-based movement capture (IMC) system, during a sports-specific field movement test, by comparing the results to those from a standard OSSCA modelling approach using marker-based optical motion capture (MMC). The identified systematic and random errors were contextualised by defining a benchmark for differences. This benchmark was generated by comparing outcomes from OSSCA to another commonly used MMC-based modelling approach, Plug-In Gait. If differences between IMC and MMC were no larger than differences between OSSCA and Plug-In Gait, we concluded that the accuracy/precision of the IMC system for the relevant variable was acceptable, i.e., within the accepted error resulting from differences in model assumptions. Joint kinematics in the sagittal plane demonstrated very high correlations, except for the trunk segment (high), with all mean RMSDs smaller than 4.44 degrees (7% of range), while mean bias ranged from 4.11 degrees (knee) to 16.60 degrees (pelvis), with all joint angles over-estimated by the IMC approach. Only the knee bias was within the benchmark (it should be noted that trunk and pelvis had no benchmark for any comparison). The hip joint was for more than half of the observations within in the benchmark. RMSD and the correlation of the knee and hip were within the benchmark (Figure 4 and Figure 7). For the frontal plane, correlations ranged from moderate (ankle and knee) to high (hip, pelvis and trunk). The observed mean RMSDs were below 3 degrees (13% of range), except for the ankle (RMSD = 9.31). The mean bias ranged from −1.41 degrees (ankle) to 9.40 degrees (knee). Compared to the benchmark, the knee bias, RMSD and correlations were within the benchmark (Figure 5 and Figure 8). For the transverse plane, correlations were very high for the pelvis and trunk, high for the hip and moderate for the ankle and knee. The ankle, knee and hip bias and RMSD were within the benchmark, while ankle and knee correlations were below the benchmark (Figure 6 and Figure 9). The observed RMSDs were between 2.86 and 6.09 degrees (14% of range), with a mean bias between −1.63 (trunk) to 9.91 (hip) degrees. For CoM measures, correlations were very high with RMSD of 0.05 and 0.07 m (2 and 4%), with a mean bias of 0.01 and −0.07 m, for the left-right and forward-backward direction. 

Studies that have previously compared outcomes of MMC and IMC systems have reported similar findings. Bolink et al. (2015) validated pelvis kinematics from an IMC motion capture system, by comparing to MMC-derived outcomes and reported similar magnitudes of RMSD (2.70 to 8.89 and 2.68 to 4.44) and r*^2^* (0.72 to 0.88) for pelvis sagittal and frontal plane angles during gait, sit-to-stand transfers and block step up transfers [4]. Lachaine et al. (2017) used an improved marker sensor alignment technique and attached optical markers on top of the IMC sensors, and reported much lower bias (<2.5), but similar RMSD (<3.0) in lower limb kinematics within an ergonomics experiment than observed in this study. This configuration, however, is not feasible for assessment of the highly-dynamic movements used in athlete field testing because of the physical restrictions on the participant resulting from sensor placement [6]. Another study, using the same IMC as used in this study, reported biases higher than those observed in this study (possibly due to model offset) and the validity was very high in the sagittal plane for hip, knee, and ankle joint angles in all three tasks. It was acceptable in frontal and transverse planes during squat and jump activity across joints due to high correlations [5]. In contrast to other studies discussed in this paragraph, Koska et al., [21] reported considerable systematic and random disagreement in ankle kinematics during running, which is in line with the findings observed in this study.

All kinematic variables, calculated from both MMC and IMC systems, inevitably contain some degree of error relative to the ‘true’ kinematics of the body. These errors result from a combination of marker placement inaccuracy, soft tissue artefacts, sensor output inaccuracy and the assumptions of the model used to translate sensor outputs into joint-level kinematics. Some lead to systematic offsets from the true joint angles (bias), whilst some result in random between-trial variation in these offsets. While, MMC techniques are considered to be the ‘gold standard’ for movement analysis, the outcome variables are highly sensitive to the locations of the markers placed on the body. Contemporary modelling approaches attempt to minimise this effect by defining joints and planes, based on movement, rather than simply marker positions (similar to [10,11,12]). However, they are nevertheless, still reliant on the accurate and consistent placement of calibration markers, whilst conventional widely-used modelling approaches rely wholly on marker positioning to define joint centres and tri-planar segment angles. Errors or inconsistencies (e.g., due to differences in thickness of subcutaneous fat) in the placement of markers relative to target anatomical landmarks is thus the single greatest contributor to measurement variability in contemporary clinical gait analysis [22]. It affects the kinematic measures by up to 15 degrees in the sagittal plane, 15 degrees in the transversal plane and 17 degrees in the transverse plane [22,23,24,25,26]. In interpreting the accuracy and precision found for the IMC system in this study, we compared the bias and RMSD to those previously reported for MMC-based systems, resulting from marker placement error within, and between, assessors (Table 2). The biases observed for the ankle, knee and hip, with the exception of hip sagittal plane angle, are within the published error ranges, resulting from MMC marker placement. However, when considering the benchmark generated in this study all three planes of the ankle (as in [21]), pelvis tilt and hip flexion all carry an error that cannot be explained by the effect of marker placement systematic error.

The increased ankle error may be explained by the fact that the foot segment’s orientation is modelled from a single IMC sensor, which is attached to the shoe tongue, whilst the MMC approach uses 3 markers (or more; 7 markers in [22]) to define the segment. As such, the IMC model may be more sensitive to movement artefacts caused by deformation of the shoe material. Errors could potentially be reduced by developing IMC solutions that rely on more than one sensor to define the orientation of the foot segment. However, this might not resolve the disagreement between MMC and IMC, as the MMC system is also sensitive to marker movement: Deformation of shoe/feet on the metatarsal can result in erroneously moved markers, and hence, errors in calculated frontal plane angles. Similarly to the ankle joint, the description of the pelvis could potentially be improved using a model using 3 pelvis sensors. The IMC system could determine pelvis tilt/drop, by measuring their offset in the vertical/horizontal axis, which should reduce disagreements in pelvis kinematics and their propagation into the hip angle (Figure 4). While the mean values of bias and RMSD for hip abduction angle are within the benchmark range, we note that the upper quartiles are consistently outside the benchmark. These errors may be due to differences in the anatomical model and other processing steps, different effects of movement artefact/soft tissue movement filters [8,27], the fact that sensors were attached to different locations, or differences in the identification of joint centres and axes between the MVN Fusion Engine and OSSCA model. Further, the findings highlight the importance in maintaining consistency in the use of one model regardless of the motion capture system (both the IMC and PiG demonstrated considerable bias and poor correlations). This needs to be considered when comparing motion analysis studies. 

The true systematic error of the IMC system itself is probably best described by the correlation and RMSD of the pelvis angles, as these are the only kinematic measures that are computed by a single sensor. Correlations between the IMC and optical motion capture were very high for pelvis tilt and rotation (r*^2^* > = 0.84) and high for pelvic drop (r*^2^* = 0.79), while RMSDs were below 3 degree. As such, the maximal systematic error should not be above 6 degrees (joint angles = proximal segment – distal segment). Errors above this threshold are likely to be random errors.

In term of the length of data capture, the accuracy and precision seem to not be impacted with a duration of 20 s, as errors did not vary across the different phases of the test.

A limitation of this study is that temporal properties (e.g., impact phase, mid-stance or toe off) of the running, change of direction or hopping cycles were not examined. Errors are likely to vary at different times, which was also highlighted by the reference trail comparison as most IMC (except rotation) measures examined were within the benchmark. This is in line with previous studies [25,28].

## 5. Conclusions

The findings of this study demonstrated that IMC sensors can be used to capture joint kinematics during an athletic field test with error levels, comparable to those introduced by modelling assumptions and marker placement variability within marker-based optical motion capture systems. However, ankle kinematics and sagittal plane hip and pelvis angles, demonstrated systematic biases and considerable random error, which likely result from sensor movement artefacts, as well as the model used to define the foot segment. Caution should be used in the interpretation of these metrics. Future development of IMC solutions, involving multiple sensors on the pelvis and foot segments, would be expected to improve accuracy and precision. Further, the findings highlight the importance of the consistency in using one model, regardless of the motion capture system used as the effect of the selected model resulted in considerable bias and poor correlations.

## Figures and Tables

**Figure 1 sensors-20-00831-f001:**
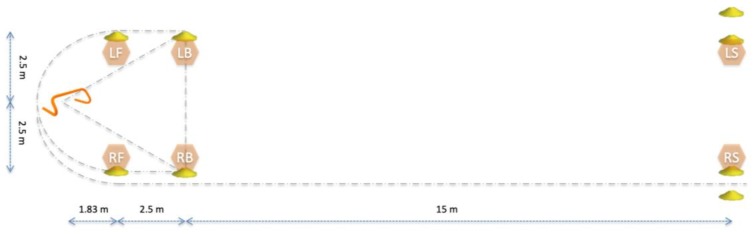
Illustration of the field test, which can be started from the right or left side of the orange hurdle (far left). For tests starting on the right side of the hurdle, the athlete stands on the right limb with hands on hips and hops over the hurdle and back (start is triggered by ground contact of the hop over the hurdle). The athlete then runs to cone RB over LB (performing a 110-degree change of direction), touches the cone RB and runs backwards to cone LB. Following this, the athlete turns around and runs around the hurdle to the inside of cone RB (performing a 110-degree change of direction at the hurdle) and performs a run clockwise around cone RB (i.e., a 180-degree turn). After this, the athlete performs a curved run to LB around RF, hurdle and LF, and performs a full stop facing outwards before performing another curve run to RS around LF, hurdle, RF and RB. When crossing RS the test is completed.

**Figure 2 sensors-20-00831-f002:**
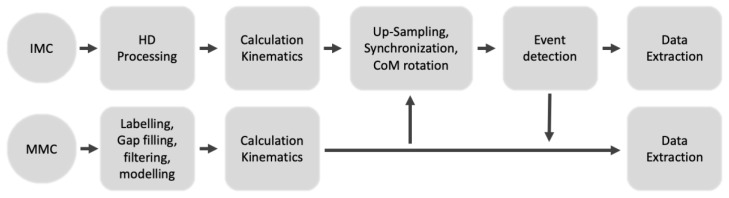
Flowchart of processing steps in the study.

**Figure 3 sensors-20-00831-f003:**
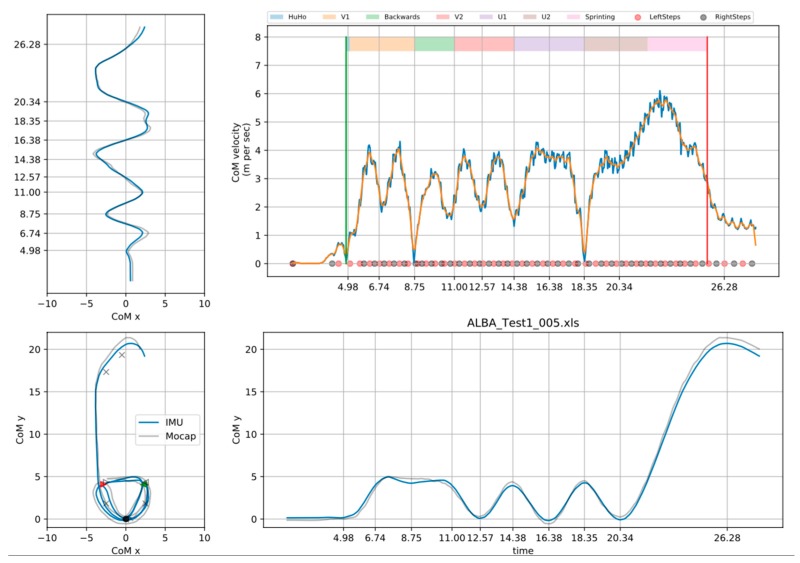
Illustration of key measures during the event registration.

**Figure 4 sensors-20-00831-f004:**
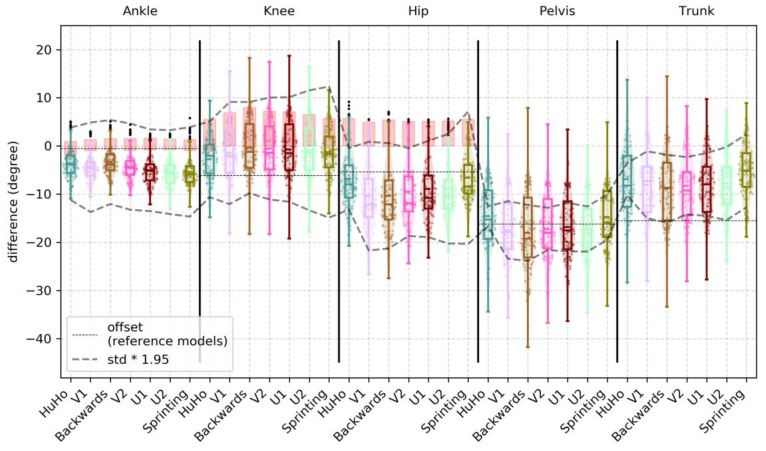
Illustration of the bias observed in the sagittal plane kinematics of the ankle, knee, hip, pelvis and trunk within every phase examined, where each point represents a trial. The red boxes in ankle, knee and hip illustrate the bias of PiG when compared to OSSCA.

**Figure 5 sensors-20-00831-f005:**
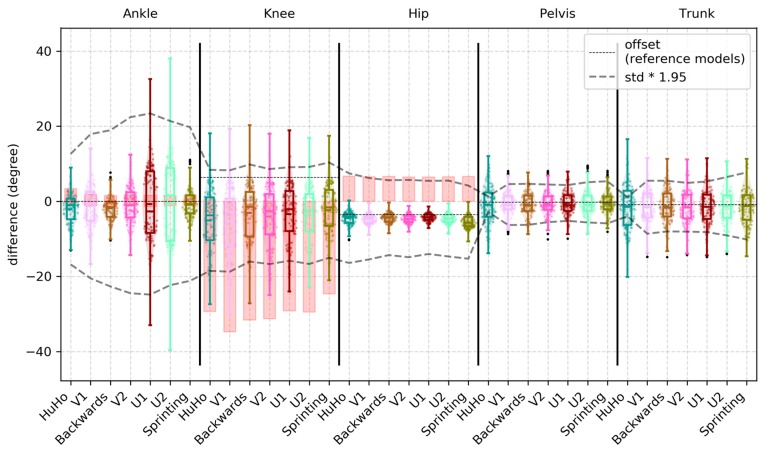
Illustration of the bias observed in the frontal plane kinematics of the ankle, knee, hip, pelvis and trunk within every phase examined, where each point represents a trial. The red boxes in ankle, knee and hip illustrate the bias of PiG, compared to OSSCA.

**Figure 6 sensors-20-00831-f006:**
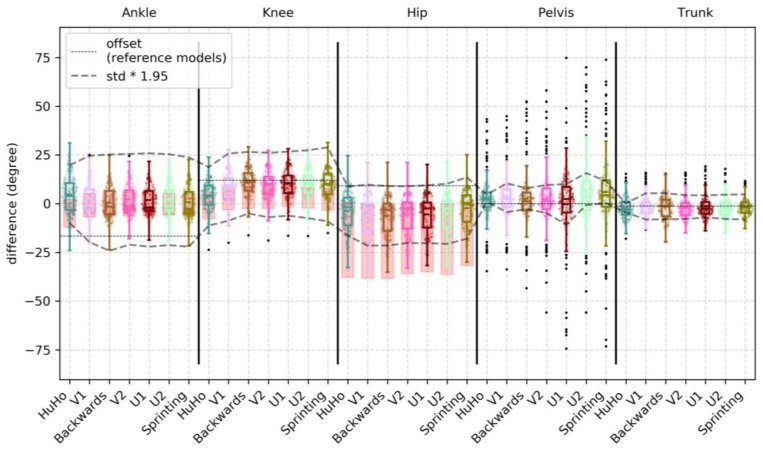
Illustration of the bias observed in the transverse plane kinematics of the ankle, knee, hip, pelvis and trunk within every phase examined, where each point represents a trial. The red boxes in ankle, knee and hip illustrate the bias of PiG, when compared to OSSCA.

**Figure 7 sensors-20-00831-f007:**
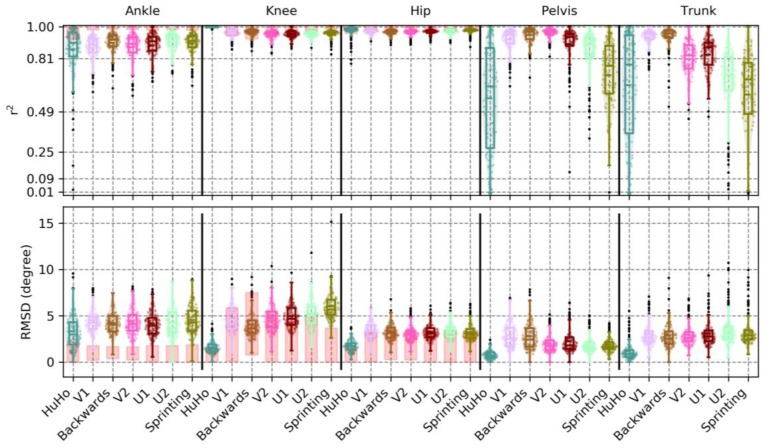
Illustration of the r*^2^* and RMSD observed in the sagittal plane kinematics of the ankle, knee, hip, pelvis and trunk within every phase examined, where each point represents a trial. The red boxes in ankle, knee and hip illustrate the r*^2^* and RMSD of PiG when compared to OSSCA.

**Figure 8 sensors-20-00831-f008:**
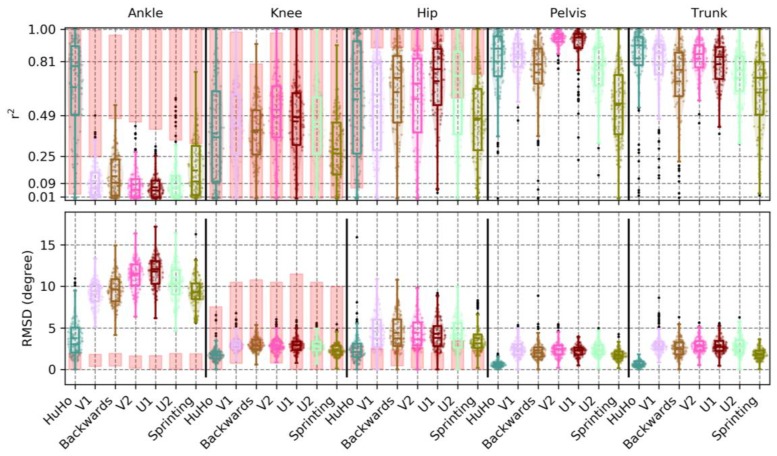
Illustration of the r*^2^* and RMSD, observed in the frontal plane kinematics of the ankle, knee, hip, pelvis and trunk, within every phase examined, where each point represents a trial. The red boxes in ankle, knee and hip illustrate the r*^2^* and RMSD of PiG when compared to OSSCA.

**Figure 9 sensors-20-00831-f009:**
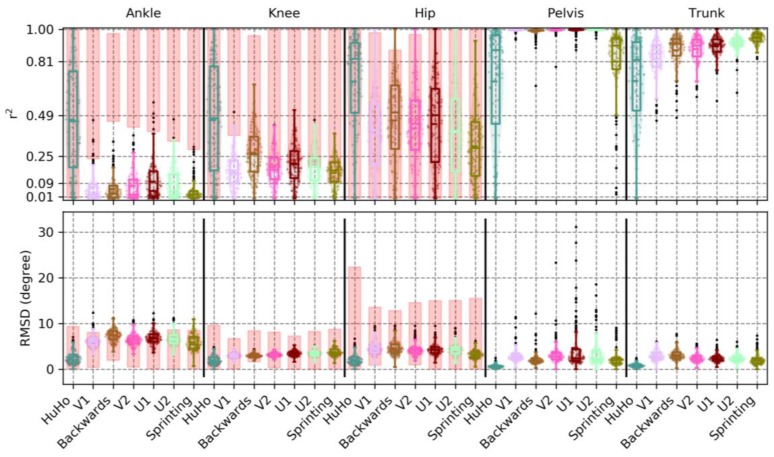
The r*^2^* and RMSD observed in the transversal plane kinematics of the ankle, knee, hip, pelvis and trunk within every phase examined, where each point represents a trial. The red boxes in ankle, knee and hip illustrate the r*^2^* and RMSD of PiG when compared to OSSCA.

**Table 1 sensors-20-00831-t001:** Description of completion times.

			Completion Time						
Subject	Session	Samples	Mean	Std	Best	Worst	25%	50%	75%
S1	Test1	10	18.336	0.733	17.210	19.290	17.784	18.320	18.910
	Test2	10	19.276	0.563	18.620	20.080	18.806	19.188	19.783
S2	Test1	10	20.934	0.485	20.205	21.665	20.591	20.883	21.299
	Test2	10	21.417	0.274	21.040	21.915	21.249	21.430	21.525
S3	Test1	10	20.576	0.419	19.730	21.195	20.323	20.663	20.826
	Test2	10	21.695	0.908	20.845	23.945	21.153	21.428	21.714
S4	Test1	10	20.236	0.347	19.860	20.935	19.960	20.115	20.325
	Test2	8	20.685	0.217	20.340	21.075	20.619	20.658	20.756
S5	Test1	9	18.627	0.380	18.340	19.360	18.351	18.493	18.653
	Test2	10	18.661	0.285	18.135	19.095	18.536	18.723	18.833
S6	Test1	10	18.237	0.217	18.075	18.765	18.100	18.135	18.325
	Test2	10	17.593	0.307	17.085	17.935	17.445	17.633	17.819

**Table 2 sensors-20-00831-t002:** Descriptive of changes/differences in kinematics measures caused due to marker placement and inter-assessor of previous studies as well as the bias and RMSD of the IMC systems error.

		Ankle			Knee			Hip			Pelvis		
		fle.	abd.	rot.	fle.	abd.	rot.	fle.	abd.	rot.	fle.	abd.	rot.
Marker error													
Cockcroft [23]	bias	-	-	-	4	−6	-	-	-	−17	-	-	-
Szczerbik [22] ^b^	bias	15	-	22	10	15	-	6	-	11	-	-	-
Groen [24] ^VCM^	RMSD	~2	~3	~9	~4	~6	~8	~3	~2	~9	~1 ^c^	~1 ^c^	~1 ^c^
Groen [24] ^OLGA^	RMSD	~2	~1	~8	~3	~2	~6	~2	~3	~6	~1 ^c^	~2 ^c^	~1 ^c^
McFadden [25]	bias	-	~1	~6	~4	~5	~5	-	-	~5	-	-	-
Inter-assessor error													
McGinley [26] ^a^	Std	2	-	-	3	2	5	4	2	5	3	2	2
Study findings													
bias		4	1	3	4	9	6	12	8	9	16	1	3
(95% limit)		(13)	(22)	(29)	(14)	(20)	(24)	(22)	(17)	(28)	(21)	(5)	(10)
RMSD		4	9	6	4	3	3	3	4	5	2	2	3

^a^ average of estimated picture readings. ^b^ worst case. ^c^ angle not affected by marker placement.

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
