# Peer review of "Agreement between Inertia and Optical Based Motion Capture during the VU-Return-to-Play- Field-Test"

_sensors, 2020, doi:10.3390/s20030831_

Round 1

Reviewer 1 Report

Thank you for submitting this work assessing the agreement between IMC and optical marker motion capture. This is an important topic to gauge the impact on measurement accuracy with increased uptake of IMC for sports movement analyses.

Here are my questions/comments on the article:

Were there any initial measurement offsets in segment angles in the calibration poses for IMC and MMC? How were calibration errors between the two systems minimised before the experiments? After calibration of both systems are you able to compare the joint positions of both the MVN model and OSSCA model and are the kinematic model the same (segment lengths, positions and initial orientations)? Were the poses of the body examined in a visualisation for the cases where OSSCA and PiG differences were high? Is it possible that agreement between PiG and IMC could be higher than OSSCA and IMC? In discussing the significance of the error or biases in sensors it would help to indicate examples where this could impact the conclusions of a study where MMC is used but IMC was another option for taking measurements. It would seem to me that some of the errors are large but then so are the benchmark differences between OSSCA and PiG in some cases. In other MMC studies how are the differences between OSSCA and PiG dealt with if at all? If there is commonly large bias this would be a major problem for other studies. What is the registered trademark VU and is there a reference for this? In preparing the data a time index was derived to align the signals however there would be some uncertainty in this. Is it possible that small timing differences can affect the agreement particularly with faster movements? If you shift the time index how does this affect the error?

Minor revision comments:

-In line 199: grammar error

-In line 280 and 289: "repect" -> "respect"

Author Response

Reviewer 1: Thank you for submitting this work assessing the agreement between IMC and optical marker motion capture. This is an important topic to gauge the impact on measurement accuracy with increased uptake of IMC for sports movement analyses.

Answer:   We would like to thank the reviewer for the suggestion given and questions raised. We fell that the corresponding changes have improved the manuscript.

R1 – Q1: Were there any initial measurement offsets in segment angles in the calibration poses for IMC and MMC? How were calibration errors between the two systems minimised before the experiments?

Answer:   Yes there were. These have not been reported previously but have now been added to the findings and discussion. See lines 211, 241-244, 252-255 and 271-274.

R1 – Q2: After calibration of both systems are you able to compare the joint positions of both the MVN model and OSSCA model and are the kinematic model the same (segment lengths, positions and initial orientations)? Were the poses of the body examined in a visualisation for the cases where OSSCA and PiG differences were high? Is it possible that agreement between PiG and IMC could be higher than OSSCA and IMC?

Answer:   Yes the visual assessment between PiG and OSSCA does illustrate differences in joint centre position / segment lengths and yes in some cases the differences between PiG and IMU would have been smaller. However while the reviewer raised an interesting point, we did not comment on this as we feel this wouldn’t aid towards the clarity of the paper.

R1 – Q3: In discussing the significance of the error or biases in sensors it would help to indicate examples where this could impact the conclusions of a study where MMC is used but IMC was another option for taking measurements. It would seem to me that some of the errors are large but then so are the benchmark differences between OSSCA and PiG in some cases. In other MMC studies how are the differences between OSSCA and PiG dealt with if at all? If there is commonly large bias this would be a major problem for other studies.

Answer:   The reviewer is right, the large differences between outputs from different models such as OSSCA and PiG are a problem that is widely known in the biomechanics literature. A variety of approaches are being developed and tested by the research community, both to quantify differences between models and to standardise reporting guidelines, but current best practice is (1) to use extreme caution drawing inferences from comparisons between studies that used different models; and (b) to always use a consistent model when tracking changes over time The reviewer raised a very important point here, which we have not discussed in full in the previous version of the paper. Hence we added a few thoughts in relation to this manner to the paper (lines 410 to 413).

R1 – Q4: What is the registered trademark VU and is there a reference for this?

Answer:   We have now included the information of the trademark holder.

R1 – Q5: In preparing the data a time index was derived to align the signals however there would be some uncertainty in this. Is it possible that small timing differences can affect the agreement particularly with faster movements? If you shift the time index how does this affect the error?

Answer:   The reviewer is right. To examine this we performed an additional experiments to determine the effect of possible misalignments by moving the along the time axis. Findings reveal that the alignment if optimized for high correlations and low bias and bias limits. See an example of changes in magnitude in respect to different time alignments (table1, 2 and 3 below). Changes in time alignment resulted in little differences in respect to bias but large differences in bias limits and correlations. The chosen performed time alignment resulted in best measures

Table 1:      Illustration of mean bias changes in responds to different signal alignments

frames moved

-10

-5

-2

0

+2

+5

+10

Hip Angles fle

10.78

10.77

10.77

10.76

10.75

10.74

10.73

Hip Angles abd

13.07

13.06

13.06

13.06

13.06

13.06

13.05

Hip Angles rot

-4.22

-4.21

-4.20

-4.20

-4.19

-4.19

-4.18

Knee Angles fle

7.94

7.94

7.94

7.94

7.94

7.93

7.92

Knee Angles abd

-12.72

-12.72

-12.72

-12.73

-12.72

-12.72

-12.71

Knee Angles rot

-0.35

-0.34

-0.34

-0.34

-0.34

-0.33

-0.32

Ankle Angles fle

-0.82

-0.83

-0.84

-0.84

-0.85

-0.85

-0.86

Ankle Angles abd

0.13

0.13

0.13

0.13

0.13

0.13

0.13

Ankle Angles rot

-23.91

-23.92

-23.93

-23.93

-23.94

-23.94

-23.96

CoM x

0.00

0.00

0.00

0.00

0.00

0.00

0.00

CoM y

0.02

0.01

0.00

0.00

0.00

-0.01

-0.02

Table 2:      Illustration of bias limits changes in responds to different signal alignments

frames moved

-10

-5

-2

0

+2

+5

+10

Hip Angles fle

24.44

16.31

12.79

11.78

12.41

15.59

23.57

Hip Angles abd

11.73

11.32

11.34

11.50

11.78

12.38

13.72

Hip Angles rot

14.46

14.03

14.02

14.18

14.48

15.14

16.53

Knee Angles fle

37.91

22.37

13.12

8.22

7.69

14.97

30.44

Knee Angles abd

7.11

6.60

6.29

6.15

6.11

6.18

6.46

Knee Angles rot

15.97

14.91

14.12

13.74

13.57

13.73

14.59

Ankle Angles fle

24.78

16.27

10.93

8.08

7.12

10.16

18.76

Ankle Angles abd

25.82

26.19

26.29

26.29

26.25

26.10

25.68

Ankle Angles rot

31.74

32.69

33.11

33.22

33.16

32.78

31.73

CoM x

0.35

0.28

0.25

0.24

0.24

0.24

0.27

CoM y

1.09

0.79

0.54

0.27

0.53

0.78

1.08

Table 3:      Illustration of correlation changes in responds to different signal alignments

-10

-5

-2

0

+2

+5

+10

Hip Angles fle

1.00

1.00

1.00

1.00

1.00

1.00

1.00

Hip Angles abd

0.87

0.95

0.98

0.98

0.98

0.96

0.88

Hip Angles rot

0.64

0.66

0.66

0.65

0.63

0.60

0.50

Knee Angles fle

0.51

0.55

0.55

0.53

0.51

0.45

0.33

Knee Angles abd

0.83

0.94

0.98

0.99

0.99

0.97

0.89

Knee Angles rot

0.30

0.40

0.46

0.48

0.49

0.48

0.44

Ankle Angles fle

-0.21

-0.03

0.10

0.16

0.18

0.16

0.02

Ankle Angles abd

0.67

0.86

0.94

0.97

0.97

0.95

0.81

Ankle Angles rot

-0.26

-0.34

-0.37

-0.37

-0.36

-0.32

-0.22

CoM x

-0.33

-0.44

-0.50

-0.51

-0.51

-0.46

-0.34

CoM y

1.00

1.00

1.00

1.00

1.00

1.00

1.00

R1 – Q6: Line 199: grammar error

Answer:   The sentence “The resulting differences in calculated kinematics can be considerable and it is thus not thus possible to compare IMC-derived kinematics against a single universal MMC-based ‘gold standard’” was change to “This can result in considerable differences in outcome measures between different models and it is therefore not possible to compare IMC-derived kinematics against a single universal MMC-based ‘gold standard’.

R1 – Q7: Line 280 and 289: "repect" -> "respect"

Answer:          These two spelling errors have been corrected. See line 291 and 300.

Reviewer 2 Report

This is an interesting manuscript that reports validity of inertial sensors during a sports movement test. The methods look sound, and results and conclusion seem reasonable. I only have some minor comments.

Title: please write out IMU. Also, revise VU-Return-to-Play-Field-Test to a general term (e.g., sports-specific movement test) if VU-Return-to-Play-Field-Test is not super well-known and established test.

Abstract: Please add discussion and conclusion. Currently, the authors just reported some results but did not really have discussion or conclusion.

Line 12: Please add "system" to the "motion capture".

Lines 14-15: Please clarify this sentence.

Methods: So, if this is a field test, how did the authors handle possible distractions to the motion capture cameras (e.g., wind, other small vibrations), as this was discussed as a weakness of motion capture system in the introduction.

Line 96: Please describe exactly how many cameras were used.

Line 100: IMU was not defined previously. Please write out and define abbreviation. In addition, throughout the manuscript, be consistent with the term either IMU or IMC, since they basically mean the same thing.

Line 112: Abbreviate Plug-in-Gait here. The authors used this term again later in the manuscript but they start abbreviating there. This is the first appearance of PIG.

Line 115: "static", I think quotation mark is not necessary here. Also, I think a different term like "reference trial" sounds more appropriate because range of motion and walking trial are not considered static. But I will leave this up to the authors' preference.

Table 1: To me, this looks more like a figure. So change this to "Figure 1".

Discussion: In the Discussion section, the authors introduced a new table "Table 3". It is very odd to see a newly introduced table in Discussion section. Please report this in Results section, and then discuss it in this section if necessary.

Author Response

Reviewer 2: This is an interesting manuscript that reports validity of inertial sensors during a sports movement test. The methods look sound, and results and conclusion seem reasonable. I only have some minor comments.

Answer:   We would like to thanks the reviewer for the interest and suggestion made to the submitted manuscript.

R2 – Q1: Title: please write out IMU. Also, revise VU-Return-to-Play- Field-Test to a general term (e.g., sports-specific movement test) if VU-Return-to-Play-Field-Test is not super well-known and established test.

Answer:   We have change the title to: Agreement Between an Inertia and Optical Based Motion Capture During the VU-Return-to-Play-Field-Test. We are still using the VU term as we clearly state that VU is a return to play test in the title.

R2 – Q2 / Abstract: Please add discussion and conclusion. Currently, the authors just reported some results but did not really have discussion or conclusion.

Answer:   We have added a brief conclusion to the abstract.

R2 – Q3: Line 12: Please add "system" to the "motion capture".

Answer:   We have added "system" to the "motion capture". See line 14.

R2 – Q4: Lines 14-15: Please clarify this sentence.

Answer:   We have altered the corresponding sentence to add clarity. It now reads: “ While kinematics from the IMC in sagittal plane demonstrated correlations (r2) between 0.76 and 0.98 with root mean square differences (RMSD) < 5, while only the knee bias was within the benchmark”.

R2 – Q5: Methods: So, if this is a field test, how did the authors handle possible distractions to the motion capture cameras (e.g., wind, other small vibrations), as this was discussed as a weakness of motion capture system in the introduction.

Answer:   The experiments were performed indoor and cameras were installed in scaffolding structures. We altered the manuscript to clearly state this. See Line 94-98.

R2 – Q6: Line 96: Please describe exactly how many cameras were used.

Answer:   In the following sentence we state that we used 70 cameras capturing at 200 Hz during the test. However, we have altered the corresponding sentences to make this more clear.

R2 – Q7: Line 100: IMU was not defined previously. Please write out and define abbreviation. In addition, throughout the manuscript, be consistent with the term either IMU or IMC, since they basically mean the same thing.

Answer:   We have altered the text throughout the main body to address this comment and use now only IMC.

R2 – Q8: Line 112: Abbreviate Plug-in-Gait here. The authors used this term again later in the manuscript but they start abbreviating there. This is the first appearance of PIG.

Answer:   We have altered the text to address this comment.

R2 – Q9: Line 115: "static", I think quotation mark is not necessary here. Also, I think a different term like "reference trial" sounds more appropriate because range of motion and walking trial are not considered static. But I will leave this up to the authors' preference.

Answer:   We have altered the term ‘static’ to ‘reference’ throughout the manuscript.

R2 – Q10: Table 1: To me, this looks more like a figure. So change this to "Figure 1".

Answer:   We have change the table to a figure.

R2 – Q11: Discussion: In the Discussion section, the authors introduced a new table "Table 3". It is very odd to see a newly introduced table in Discussion section. Please report this in Results section, and then discuss it in this section if necessary.

Answer:   We did not move the table as it does not correspond to the findings of this study and is rather an overview of previous research that is needed for the discussion.
